# Effects of Intermittent Energy Restriction Alone and in Combination with Sprint Interval Training on Body Composition and Cardiometabolic Biomarkers in Individuals with Overweight and Obesity

**DOI:** 10.3390/ijerph19137969

**Published:** 2022-06-29

**Authors:** Matthew B. Cooke, William Deasy, Elya J. Ritenis, Robin A. Wilson, Christos G. Stathis

**Affiliations:** 1College of Health and Biomedicine, Victoria University, Melbourne, VIC 3000, Australia; w.deasy@cqu.edu.au (W.D.); robin.wilson@nyulangone.org (R.A.W.); 2Department of Health Sciences and Biostatistics, Swinburne University of Technology, Melbourne, VIC 3122, Australia; eritenis@swin.edu.au; 3Australian Institute for Musculoskeletal Science (AIMSS), Western Health, Melbourne, VIC 3021, Australia; 4College of Clinical Sciences, Central Queensland University, Rockhampton, QLD 4701, Australia; 5Institute for Health and Sport, Victoria University, Melbourne, VIC 8001, Australia

**Keywords:** intermittent fasting, sprint exercise, metabolic, body composition, cardiorespiratory fitness

## Abstract

The popularity of intermittent fasting (IF) and high intensity (sprint) interval training (SIT) has increased in recent years amongst the general public due to their purported health benefits and feasibility of incorporation into daily life. The number of scientific studies investigating these strategies has also increased, however, very few have examined the combined effects, especially on body composition and cardiometabolic biomarkers, which is the primary aim of this investigation. A total of thirty-four male and female participants (age: 35.4 ± 8.4 y, body mass index (BMI): 31.3 ± 3.5 kg/m^2^, aerobic capacity (VO_2peak_) 27.7 ± 7.0 mL·kg^−1^·min^−1^) were randomized into one of three 16-week interventions: (1) 5:2 IF (2 non-consecutive days of fasting per week, 5 days on *ad libitum* eating), (2) supervised SIT (3 bouts per week of 20s cycling at 150% VO_2peak_ followed by 40 s of active rest, total 10 min duration), and (3) a combination of both interventions. Body composition, haemodynamic and VO_2peak_ were measured at 0, 8 and 16 weeks. Blood samples were also taken and analysed for lipid profiles and markers of glucose regulation. Both IF and IF/SIT significantly decreased body weight, fat mass and visceral fat compared to SIT only (*p* < 0.05), with no significant differences between diet and diet + exercise combined. The effects of diet and/or exercise on cardiometabolic biomarkers were mixed. Only exercise alone or with IF significantly increased cardiorespiratory fitness. The results suggest that energy restriction was the main driver of body composition enhancement, with little effect from the low volume SIT. Conversely, to achieve benefits in cardiorespiratory fitness, exercise is required.

## 1. Introduction

Obesity rates continue to rise worldwide, with one-in-five children and adolescents classified as overweight [1]. Excessive weight is a risk factor for many of the world’s leading causes of death, including diabetes, heart disease, stroke, and various types of cancer and as a result, is responsible for over four million premature deaths a year [1]. Higher energy intake and lower energy expenditure leading to energy imbalance has been identified as the predominate contributor to the aetiology of obesity. However, it is understood that it is not as simple as “energy in versus energy out” and may involve many complex factors including: genetics, changes in the epigenome, dysbiosis of gut microbiota, environmental, socio-economic, and psychological [2]. For adults with obesity, weight reduction through continuous energy restriction and/or increased physical activity is usually recommended and has shown to be an important component of weight loss and weight maintenance programs [3]. In addition to weight loss, exercise and/or calorie restriction can provide other benefits by improving physical function and health-related quality of life, including, but not limited to, reductions in blood pressure [4], fasting insulin levels, total cholesterol, triglycerides and low-density lipoproteins [5], and inflammation [6]. 

Though continuous energy restriction is commonly prescribed by health/nutritional professionals and is effective in the short-term period, long-term adherence with daily energy restriction is often poor [7]. Over recent years, intermittent fasting (IF) has emerged as a promising alternative to continuous/traditional calorie restriction [8]. Defined generally by alternating periods of fasting with unrestricted eating, IF can fall under six different approaches: religious fasting, alternate-day fasting (ADF), alternate-day modified fasting (ADMF), twice-weekly fasting (i.e., 5:2 diet), modified periodic fasting usually called ‘fasting-mimicking diet’, and time-restricted eating (TRE) [9]. While some reports suggest IF facilitates greater adherence and potentially long-term weight management success [10], there is currently no clear consensus. Regardless, the short-to medium-term health benefits of IF diets are many, especially within health conditions such as obesity, diabetes, and cardiovascular diseases [11]. These benefits are usually driven by the weight loss, however, preliminary studies suggest that IF might also be beneficial in the absence of weight loss compared to other types of energy restricted diets [12], therefore providing health benefits to normal-weighted and/or weight stable individuals.

An unwanted consequence of most energy restricted diets is the loss of muscle mass during the weight loss phase [13]. Reductions in muscle mass (which is the major component of lean body mass (LBM)) is undesirable as it is known to impair physical functionality, cardiometabolic health and may place an individual at higher risk for weight regain [14,15,16]. Some studies suggest that IF regimes maybe more protective of LBM compared with continuous energy restriction [17,18], while others have reported greater LBM loss with IF regimens [14,15,19]. Regardless, employing interventions that can promote muscle mass growth such as resistance training which can fully or partially attenuate this unwanted side effect is important when undertaking any energy-restricted diet. While few studies have recently examined the combined effects of IF (and variations of) with resistance exercise [20], a paucity of studies have examined the concurrent benefits of high intensity/sprint interval exercise. 

Interval training, in the form of high-intensity interval training (HIIT) and sprint interval training (SIT) are promoted as superior and time efficient exercise models for enhancing body composition (i.e., decreasing body and fat mass) and reducing metabolic risk factors for chronic diseases compared to moderate intensity continuous training (MICT). Interval training refers to brief bouts of vigorous intensity exercise interspersed by relatively longer bouts of low intensity active recovery or passive recovery [21,22], with evidence suggesting that as little as 10 min of SIT can improve metabolic health and aerobic capacity [23]. We recently examined the independent and combined effects of IF and HIIT on anthropometric and metabolic health parameters in a model of diet-induced obesity [24]. Our study found that IF combined with HIIT induced superior effects on body composition and lipid profiles compared to either diet or exercise intervention alone [24]. These findings were also supported by both sexual-dimorphic and tissue-specific adaptations within the hypothalamus, adipose tissue and skeletal muscle [25]. 

The purpose of the present study was to confirm the aforementioned animal model findings in humans by mimicking similar IF and HIIT protocols combined and in isolation in overweight and obese males and females for 16 weeks. We hypothesised that the combination of sprint interval training (SIT) with IF may lead to greater synergistic effects on fat mass reduction, improved aerobic fitness and retention of LBM than either SIT or IF alone. Moreover, these superior body compositional and cardiorespiratory improvements will translate into greater reductions in cardiometabolic risk factors. 

## 2. Materials and Methods

### 2.1. Participants 

Participants were recruited through advertising on social media channels targeting the western regions of Melbourne, Australia. A total of 96 individuals responded to the advertisement and underwent initial screening. Only 45 were deemed eligible and were recruited for the study. Participants were eligible for inclusion if they: (i) were aged between 18–45 years; (ii) had a body mass index (BMI) of 25–35 kg/m^2^; (iii) had a body fat percentage >22% for males or >31% for females as measured via dual x-ray absorptiometry (DXA); (iv) had not followed a structured training program in the previous 6 months and; (v) had been weight stable for 3 months prior to the study (<5% weight loss or weight gain). Participants were excluded if they: (i) were smokers; (ii) had a diagnosed chronic disease or condition; (iii) were taking medications or dietary supplements deemed contraindicative to the study intervention and were unwilling to cease these for the duration of the study; (iv) were pregnant or intended to become pregnant in the following 3–4 months; and (v) were menopausal or post-menopausal. This project was approved by the Victoria University Human Research Ethics Committee (approval No. HRE 15-160) and registered with the Australian and New Zealand Clinical Trials Register (ACTRN12615001331527). 

### 2.2. Study Overview 

An overview of the participant recruitment and final numbers for analysis is presented in Figure 1. Following screening and informed consent, participants were issued a 4-day food diary to monitor dietary intake for one week prior to starting the study and completed questionnaires related to physical activity (IPAQ) to ascertain baseline dietary intake and physical activity levels. Baseline testing (Week 0) was completed over 2 days. On day 1, participants arrived at the research testing facility between 0700 and 0900 to undertake the following: (1). Assessment of body weight and waist to hip ratio, (2). Body composition scan via DXA, (3). Fasting blood sample collection and (4). Oral glucose tolerance test (OGTT). On day 2, participants arrived at the same time as the previous day and underwent cardiovascular analysis (blood pressure and pulse wave velocity) followed by aerobic capacity testing (VO_2max_). The aforementioned measures were repeated in weeks 8 and 16 of the study. Participants were required to complete weekly 4-day food logs (3 non-fasting days and 1 fasting day for both fasting groups) and activity questionnaires (IPAQ). 

### 2.3. Randomisation and Intervention Protocols

Participants were randomly allocated on a 1:1:1 basis to either 16 weeks of twice-weekly fasting (5:2) diet, three sessions a week of SIT, or a combination of the two protocols. Block randomisation was used in an attempt to match groups based on baseline BMI, age, predictive daily calorie intake and physical activity levels.

The 5:2 IF only group restricted energy intake for two non-consecutive days per week for 16 weeks, with *ad libitum* food intake for the remaining five days of the week. Fasting day intake was limited to a single 600 kilocalorie (kcal) and 500 kcal meal for men and women, respectively, as well as *ad libitum* consumption of water and unsweetened black tea. Dietary advice and sample recipes in line with current Australian nutritional guidelines [26] was provided on fasting and non-fasting days. 

The SIT group attended three supervised training sessions per week for 16 weeks consisting of 4 × 20 s work at 150% VO_2peak_ followed by 40 s of active rest at 50 watts on a magnetically braked bicycle ergometer (Lode Excalibur, Lode B.V., Groningen, The Netherlands). Workload was increased from four to six repetitions over the first 4 weeks and remained at six intervals for the study duration. Three-minutes of warm-up and warm-down (at 50 watts of resistance) before and after the exercise period were also included. Participants were required to attend a minimum of 85% of training sessions, with any participant falling below this threshold excluded from the final analysis.

Participants in the combined group (IF/SIT) followed the same dietary protocol as per the IF only group and the exercise protocol as per the SIT only group. Exercise training was undertaken on non-fasting days. 

### 2.4. Anthropometric and Fitness Assessments

#### 2.4.1. Body Weight, Height and Body Mass Index 

Height and weight was measured in a fasted state (minimum 8 h) and after voiding the bladder using a wall mounted stadiometer and bioimpedance scale (Tanita Innerscan, Kewdale, Australia). BMI was calculated using the formula BMI = kg/m^2^.

#### 2.4.2. Body Composition Assessment 

Body composition was assessed using a whole-body DXA scan according to the manufacturer’s instructions (Discovery W, Hologic Inc., Marlborough, MA, USA). Measurements for visceral fat mass, lean body mass and fat mass were recorded for each participant. 

### 2.5. Haemodynamic Assessment 

#### 2.5.1. Central and Peripheral Blood Pressure 

Participants were instructed to lie comfortably in a supine position on a table and were fitted with pneumatic blood pressure cuffs on both their upper arm and thigh. Following 10 min of rest and minimal movement, peripheral and central blood pressure was assessed using the SphygmoCor Xcel analyser (AtCor Medical, Sydney, Australia). 

#### 2.5.2. Pulse Wave Velocity 

In the supine position, the carotid pulse, sternal notch, femoral pulse and position of the thigh cuff were measured for distance between landmarks and entered into the software (XCEL version 1.2.0.7, AtCor Medical, Australia) to provide the pulse wave distance. A tonometer was then placed over the carotid pulse until a steady pulse measurement was registered by the software, causing the thigh cuff to inflate automatically. The software then compared the pulse measurements to estimate the pulse wave velocity (m/s).

### 2.6. Aerobic Capacity (VO_2_) Testing

Participants abstained from eating for an hour prior to testing. A graded exercise test using a ramped workload protocol was performed on a magnetically braked cycle ergometer (Lode Excalibur, Lode B.V., The Netherlands). Male participants began at a power output of 50 watts for three minutes after which the workload was increased incrementally by 50 watts every 3 min until 9 min, after which the load increased in 25 watts increments each minute until volitional fatigue. The female protocol varied in that the starting load was 25 watts and that all subsequent increases in load were 25 watts. Oxygen uptake (VO_2_) and respiratory exchange ratio (RER) were measured every 30 s. VO_2peak_ was determined to have been achieved upon volitional fatigue or if two of the following criteria were met: a respiratory exchange ratio ≥ 1.15, an RPE score ≥ 19 and/or a peak heart rate within ±10 beats of the participants’ age predicted maximum (determined as: 220-age). Heart rate was continuously monitored with perceived exertion (Borg scale rating of perceived exertion) assessed during each interval. 

### 2.7. Cardiometabolic Biomarker Assessment

#### 2.7.1. Serum and Plasma Markers of Metabolic and Cardiovascular Risk 

Blood was collected via catheterisation of the antecubital vein at least 72 h following the last bout of exercise and at least 8 h after the participants’ last meal. Following catheterisation, blood samples were taken before glucose ingestion (OGTT), at 30 min, 60 min and 120 min. Baseline samples (prior to glucose ingestion) were used for all blood analysis, with samples taken after glucose ingestion used for glucose area under the curve (AUC). Serum tubes were left to sit for at least 20 min, but less than 40 min, to allow clotting. The samples were then centrifuged for 12 min at 3000 RPM to separate the serum. Serum was then aliquoted and stored at −80 °C. Plasma tubes were delivered on ice directly to an external commercial lab (Dorevitch Laboratories, Melbourne, Australia) for measurement of lipids, glucose and haemoglobin A1c (HbA1c). Serum insulin levels were analysed using enzyme linked immunosorbent assays (Abcam, UK and Thermo Scientific, Waltham, MA, USA). 

#### 2.7.2. Oral Glucose Tolerance Testing

Participants consumed a drink containing 75 g of glucose following which blood samples were collected at baseline, 30 min, 60 min and 120 min. Glucose tolerance results were reported as area under the curve (AUC) values calculated using the trapezoidal method, with insulin resistance determined by homeostatic model assessment of insulin resistance (HOMA-IR) using the equation ((fasting blood glucose × fasting insulin)/22.5) [27].

### 2.8. Nutritional Intake Assessment

Participant nutritional data was collected by either a 4-day food log (for participants without smart phones or those who were not technically proficient) or using MyFitnessPal™. Participants were asked to record four days of food intake, with those in fasting groups asked to record one (only) of their fasting days to assess dietary compliance. Food diaries were analysed using Foodworks 9 software (Xyris Software, Brisbane, Australia). 

### 2.9. Physical Activity Monitoring

Participant activity levels that were outside of the study related exercise (SIT and IF/SIT groups only) were monitored using the international physical activity questionnaire (IPAQ) to assess weekly physical activity (in minutes). 

### 2.10. Statistical Analysis 

An *a priori* power analysis was carried out in G*power (Heinrich Heine University, Düsseldorf, Germany) using data sourced from a selection of similar fasting studies (α = 0.05, effect size = 0.25 and correlation of 0.7). Based on a predicted dropout rate of 30%, an estimated 45 participants were required to achieve a total of 32 participants. Intention-to-treat data was analysed using a last observation carried forward (LOCF) manner. Absolute and change from baseline data was analysed using repeated measures ANOVA to test for within-subject and between-subject effects. Male and female participants were pooled together in each group due to the small number of male participants present in each group (IF = 1, SIT = 2 and IFSIT = 3). In addition, one participant from the SIT group was removed for glucose AUC analysis due to an error in the baseline reading. Correction for multiple pairwise comparisons was carried out using Tukey’s honest significant difference (Tukey HSD) calculation. Demographic data were analysed via MANOVA to test for significant differences at baseline. Statistical analyses used SPSS version 26 (IBM, Armonk, NY, USA) applying an α value of 0.05.

## 3. Results

### 3.1. Participants 

A total of forty-five participants were recruited for this study with an overall dropout rate of 38% by the end of the 16 weeks (Figure 1). A total of 28 participants completed the study (IF = 8, IF/SIT = 11, SIT = 9) with a further six participants included in the intention-to-treat analysis (IF = 4, IF/SIT = 0, SIT = 2) giving a total of 34 participants included in the final analysis. Baseline characteristics for the 34 participants are reported in Table 1. Statistical analysis revealed a significant difference in BMI between IF and IF/SIT groups (*p* = 0.047) and a moderate trend for body weight (*p* = 0.087). No other significant differences between groups at baseline were identified. 

### 3.2. Adherence to Exercise and/or Dietary Intervention 

A total of 48 sessions were conducted for the sprint interval training, with both groups completing greater than 88% of sessions. Dietary compliance (fasting days) was also high, with both IF and IF/SIT groups recording a dietary compliance greater than 80%.

### 3.3. Anthropometric Analysis before and after Intervention 

#### 3.3.1. Body Weight 

A main effect for time (*p* < 0.001), group (*p* = 0.005) and group by time interaction (*p* = 0.004) was identified for body weight, with significant reductions observed in both fasting groups (IF: *p* = 0.024, IF/SIT: *p* = 0.015) compared to the SIT only group over 16 weeks (Figure 2A). This was also confirmed in the change (Δ) from baseline analysis, with significant reductions in body weight occurring in the IF group at both 8 weeks (*p* = 0.04) and 16 weeks (*p* = 0.001). There were also significant reductions in body weight in the IF/SIT group at 8 weeks (*p* = 0.009); albeit reductions at 16 weeks were only a trend (*p* = 0.09). Both fasting groups demonstrated significantly lower body weights at 8 weeks (IF: *p* = 0.008, IF/SIT: *p* = 0.01) and 16 weeks (IF: *p* = 0.02, IF/SIT: *p* = 0.01, Figure 2A) compared to the SIT only group. No statistical differences were identified between the fasting groups at either time point.

#### 3.3.2. Waist Circumference and Waist to Hip Ratio

A significant main effect for time was revealed for waist circumference (*p* = 0.022, Table 2), with all groups decreasing their waist circumference over the 16 weeks, albeit, only a strong trend was statistically identified for IF/SIT at 16 weeks (*p* = 0.051). No other main effects and interactions were identified for waist circumference and waist to hip ratio (Table 2).

#### 3.3.3. Body Composition

A significant main effect for time (*p* < 0.001), group (*p* = 0.008) and group by time (*p* = 0.002) was revealed for fat mass. Similar to body weight, significant reductions in fat mass were observed in both fasting groups (IF: *p* = 0.019, IF/SIT: *p* = 0.035) compared to the SIT only group over 16 weeks. This was also confirmed in the Δ from baseline analysis, with significant reductions in fat mass only occurring in the IF and IF/SIT at both 8 weeks (*p* = 0.09, *p* = 0.02; data not shown) and 16 weeks (*p* = 0.0001, *p* = 0.02). Both fasting groups demonstrated significantly lower fat mass at 16 weeks (IF: *p* = 0.004, IF/SIT: *p* = 0.02, Figure 2C) compared to the SIT only group. No statistical difference was identified between the fasting groups for either analysis.

No significant main effects for time, group or group by time were revealed for LBM. While Δ from baseline analysis revealed no significant changes within-group, between-group analysis showed a significant difference in LBM change between SIT and IF (*p* < 0.01) and SIT and IF/SIT (*p* < 0.01) at week 8 (data not shown), with LBM loss evident in both fasting groups with or without exercise compared to SIT. No significant differences between-groups were observed at week 16 (Figure 2D).

Visceral adipose tissue (VAT) analysis revealed a significant effect for time (*p* = 0.001), with a significant decrease in VAT within the IF/SIT group at 16 weeks (*p* = 0.004). No other significant main effects or interactions were identified. The Δ from baseline analysis revealed no significant changes within-group at 8 weeks, but significant reductions at 16 weeks in the IF/SIT group (*p* = 0.004). A trend was noted in the IF only group (*p* = 0.07). No significant differences between-groups were noted (Figure 2B).

### 3.4. Aerobic Capacity

A significant main effect for time (*p* = 0.001), group (*p* = 0.001) and group by time interaction (*p* = 0.002) was revealed for VO_2peak_, with a significant increase in VO_2peak_ in the SIT only (*p* = 0.03) and IF/SIT (*p* = 0.001) groups compared to IF only group. At the end of the intervention, VO_2peak_ was significantly higher in the IF/SIT group compared to the IF only group (*p* = 0.003, Figure 3). This was also confirmed in Δ from baseline analysis, with a significant increase in ΔVO_2peak_ for both SIT and IF/SIT groups at 16 weeks (*p* < 0.05), and a significant between-group difference between IF and IF/SIT groups (*p* < 0.01).

### 3.5. Haemodynamic Measures

No significant main effects for time, group or group by time were revealed for both peripheral and central blood pressure measurements and pulse wave assessment (Table 3). Similarly, Δ from baseline analysis at 16 weeks revealed no significant within-group or between-group differences.

### 3.6. Cardiometabolic Biomarkers

#### 3.6.1. Triglycerides, Total Cholesterol, LDL and HDL

No significant main effects for time, group or group by time were revealed for triglycerides, total cholesterol, LDL and HDL (Table 4). Similarly, Δ from baseline analysis at 16 weeks revealed no significant within-group or between-group differences for total cholesterol and HDL. However, a significant reduction was found for LDL levels with IF/SIT group (*p* < 0.01).

#### 3.6.2. Glucose, Markers of Glucose Tolerance and Insulin Resistance

No significant main effects for time, group or group by time were revealed for fasting glucose, HbA1c, glucose tolerance AUC or HOMA-IR (Table 4). Similarly, Δ from baseline analysis at 16 weeks revealed no significant within-group or between-group differences for fasting glucose, HbA1c, and HOMA-IR. However, a significant reduction was found for glucose tolerance AUC levels with IF/SIT group (*p* < 0.05).

### 3.7. Dietary Intake Analysis

No significant main effects for time, group or group by time were revealed for total energy intake, carbohydrate intake, fat intake or protein intake during the 16-week intervention period (Table 5). Nutritional intake on fasting days was similar between IF and IFSIT groups, with no significant differences noted.

### 3.8. Physical Activity Levels

No significant main effects for time, group or group × time were revealed for total physical activity during the 16-week intervention (Table 6). Analysis of exercise modality/intensity revealed no significant differences over time or between-groups in any form of modality/intensity.

## 4. Discussion

The primary aim of the present study was to determine if twice weekly intermittent fasting combined with high intensity exercise would provide superior benefits on weight loss and improvements in cardiometabolic biomarkers compared to either intervention alone. The results showed that intermittent fasting with or without a 3-day/week sprint exercise training program provided similar effects on body composition (i.e., body fat and lean muscle mass) in overweight and obese individuals that were predominantly metabolically healthy. However, the effects on haemodynamic measures (i.e., blood pressure) and clinically relevant markers in lipids (i.e., triglycerides and cholesterol) and glucose regulation (i.e., AUC, HOMA-IR) were mixed. A clear benefit from the addition of exercise to intermittent fasting was an improvement in aerobic capacity. Only exercise training alone and when combined with intermittent fasting improved aerobic capacity, with no change in the intermittent fasting only group. Despite such improvements in cardiorespiratory fitness, this did not translate into benefits across all exercise groups in haemodynamic and clinically relevant markers. The study findings suggest that either intermittent fasting or intermittent fasting combined with sprint exercise training is effective at inducing weight loss, specifically body fat levels. However, maintenance or improvements in cardiovascular health would require exercise training.

The benefits of intermittent fasting on body composition are well established [11,28]. While different variations of intermittent fasting exist, all involve extended periods of time with little or no nutritional intake. The 5:2 intermittent fasting protocol is a popular method which incorporates 2 days of fasting consuming only water (termed ‘zero-calorie’ fasting’) or, allowing 25% of energy needs (approximately 500 kcal per day) called ‘modified ADF’, with days of unrestricted consumption [29,30]. Findings to date reveal that 5:2 diet is effective in reducing weight (4–8% loss from baseline) over the short-term period (8–12 weeks) in men and women with obesity [31,32,33]. However, longer-term studies (i.e., 52 weeks) demonstrate no greater reductions suggesting a peak in weight loss efficacy from this type of diet around 3 months [29,32,34]. Compared to daily calorie restriction, weight loss appears to be similar in both short-term [35] and long-term trials [36]. The present study supports the previous findings demonstrating comparable reductions in weight loss (~6%) in the IF only group over the 16-week period, where reductions in total energy intake in the range of 10–30% of total kilocalories consumed per day are evident.

Reductions in fat mass are the main contributor to weight loss following energy restricted diets [37]. In the present study, reduction in fat mass was approximately ~83% (~3.5 kg) of total weight loss, which is slightly higher than other studies [38,39]. Importantly, a high proportion of fat mass loss was from visceral fat, a region of fat that is more strongly correlated with ill health compared to others (i.e., subcutaneous). This has also been shown in other studies [32,38], though a recent review suggests the majority of weight is lost through reductions in subcutaneous fat mass [29]. An unfavourable consequence of energy restriction is loss of LBM, with around 25% of body weight loss in the form of lean muscle mass [37]. In the present study, LBM loss following intermittent fasting was less (~17.5%, ~0.8 kg) which may support previous suggestions that IF leads to greater attenuation of LBM loss compared to continuous energy restriction [40], though more recent evidence suggests otherwise [41]. On the contrary, reductions in LBM between energy-restricted diets could be dependent on the percentage of weight loss, with participants who lose more weight (i.e., >10% of their body weight) likely to lose greater amounts of LBM. Indeed, in the present study, IF only group lost on average 5% of their body weight at the end of the intervention, and this is a more likely reason for the less than typical loss of LBM while undertaking an energy restricted diet. A limitation of the current study is that we did not include a continuous energy restriction only group and thus we can only speculate on such reasons.

A common strategy to minimise loss in LBM and potentially promote muscle accrual is combining the energy restricted diet with exercise training, specifically resistance training [20]. Conversely, low volume HITT has been shown in a recent meta-analysis to be non-effective at increasing LBM when compared with non-exercising control [42]. Whilst most studies reviewed tended to favour an improvement in LBM with HIIT versus moderate-intensity continuous training, this was not significant [42]. The results in the present study support these previous findings with no significant increase in LBM within the SIT only group. The combined effects of HITT and fasting on body composition have also been examined in a recent systematic review and meta-analysis [43]. Subgroup analysis found that compared to the control group (fasting alone or HIIT alone), LBM was improved in the short-term (<4 weeks) when combined with HITT, whereas long-term (>4 weeks) the opposite was found [43]. However, given the majority of ‘intermittent fasting’ interventions included in the analysis were continuous calorie restriction and not fasting for prolonged periods, it is difficult to apply their findings to the present study results. A recent study did compare the effects of 8 weeks of HITT with or without daily intermittent fasting in the form of time restricted feeding on body composition and performance [44]. Predicted estimates of muscle mass was similar between interventions and unchanged over the 10 weeks. However, given the cross-over design of the study, different training volume (~40 min vs. ~10 min) and fasting type (twice weekly vs. daily time restricted) compared to the present study, it is again difficult to directly compare to the current study results.

Therefore, to our knowledge, no human study has investigated the combined effect of sprint exercise and 5:2 intermittent fasting on LBM and our results suggest that 10 min of sprint exercise, three times a week is not sufficient to attenuate LBM loss that may occur with intermittent fasting. Interestingly, LBM loss was (non-significantly) greater in the combined group compared to intermittent fasting alone (3.1% vs. 2.5%). Given body weight loss and fat loss were comparable between groups, it is unlikely to be a negative energy balance issue, which is known to impact muscle protein metabolism [45]. On the contrary, it is possible that insufficient protein intake, which is another major determinant of training-induced changes in LBM [46,47], could be a contributing factor, with intakes in the combined group just below or at RDA (0.8 g·kg·bw per day). While all groups were, on average, lower than the recommended 1.2 to 1.9 g·kg·bw per day recommended for LBM maintenance and accrual [46,48], both SIT and IF were higher compared to IF/SIT.

Combining IF with sprint exercise training did not result in greater weight loss and/or fat mass. While some studies suggest superior improvements in weight and/or fat loss when exercise is added to IF [49,50], others have shown no improvement when diet and exercise are combined [51]. In the present study, the combined exercise and diet intervention group was only superior to SIT only. Given little change in body weight in the SIT group, it is not surprising that no additional weight loss and/or fat loss was observed in the combined group when compared to IF only. The findings in the SIT group confirm a recent meta-analysis suggesting low-volume HIIT was no better in improving body weight and fat mass when compared with a non-exercising control or moderate-intensity continuous training (MICT), which is likely a reflection of HIIT protocols characterised by low energy expenditure of ≤ 500 metabolic equivalent (MET).min per week (and associated time reduction) [42].

The effects of low volume HIIT/SIT on cardiometabolic biomarkers in overweight and obese, but otherwise healthy individuals, are equivocal with some demonstrating improvements in blood pressure and fasting blood glucose levels [52,53] while others show no significant change in blood pressure, blood lipid levels, or glucose control [54]. Total workload ranged from 1 × 4-min at 90% HRmax to 12 × 1-min at 80–90% HRmax, which are higher volumes compared to the current study (4–6 × 20 s bouts). The lack of effect from low volume SIT on cardiometabolic biomarkers in the present study is unlikely due to the timeframe, with others suggesting at least 8–12 weeks is required for HIIT to demonstrate a positive impact on physiological adaptations that improve cardiometabolic health in overweight/obese populations [55]. A possible explanation could be that most participants in the present study presented blood markers and blood pressure already within standard-range at baseline, thus reducing the likelihood of observing notable group differences [54]. In contrast, change from baseline analysis did reveal significant reductions in the IF/SIT group for the marker of glucose tolerance and LDL cholesterol levels. While others have also shown benefits when an exercise program is combined with fasting [50], given the lack of effects from SIT only, it is likely that such metabolic benefits were driven by the fasting and/or weight loss noted in both IF and IF/SIT groups. Indeed, the IF group also demonstrated reductions in glucose AUC and HOMA-IR, though they were not statistically significant.

A noticeable benefit of SIT in the present study was the cardiorespiratory fitness enhancement. Our results extend the evidence that low-volume HIIT interventions are effective for enhancing cardiorespiratory fitness, despite requiring lower energy expenditure and volume compared to MICT [42]. Gist et al. [56] demonstrated an ~ 8% improvement in VO_2max_ with SIT protocols, which was superior to non-exercising controls and equally effective to MICT in adults. Research suggests that for every increase in estimated one metabolic equivalent (MET, corresponding to oxygen consumption of 3.5 mL·kg^−1^·min^−1^), there is an associated ~15% reduction in cardiovascular disease risk [57]. Our results showed that SIT only and IF combined with SIT improved VO_2peak_ by nearly one and two METs, respectively, and thus could be providing additional benefits in disease risk reduction.

Dietary intake data revealed that all groups were in calorie deficit over the 16 weeks. While this is expected for IF and IF/SIT groups, this was not prescribed for the exercise only group. This could be in part a reflection of “participant/response bias” where participants that typically enrol into diet/exercise or weight loss trials are those already looking to lose weight and thus subconsciously reduce their calorie intake regardless of which group they are assigned too. Despite the calorie reduction, the exercise only group did not lose significant weight. Whether the dietary log information provided by participants was a true reflection of what they were eating cannot be accurately determined; which is a limitation of using such tools [58]. Physical activity outside of study training (for SIT and IFSIT groups only) was similar over the 16 weeks with no significant changes or differences between groups reported.

Our study design is not without its limitations. We wanted to replicate a ‘real-world’ environment to determine feasibility of both diet and exercise interventions. Thus, participants were asked to undertake both diet and/or exercise programs in a free-living environment where their diet and exercise outside of the study trial was not tightly controlled but guided in the form of healthy eating recommendations and instructions to maintain normal physically activities during the 16 weeks. While this doesn’t allow the benefits of either fasting and/or sprint exercise to be determined independently of weekly ‘energy balance’, it does allow us to observe any compensatory mechanisms that may occur as a result of either or both interventions which may also impact the benefits of such interventions on outcomes measured. In addition, given the small sample size in each group and low males participant numbers, we pooled together both sexes in the analysis. While there is evidence to suggest sex-specific responses to both interventions, especially in animal models [25], given the large proportion of one sex in each group and a similar number of different sexes within each group, it is unlikely such differences influenced results, but we cannot exclude this. Given the longitudinal nature of the study design, variations in menstrual cycles were not controlled for and thus any effects, if any, cannot be ruled out. Finally, the SIT protocol used in the present study was selected based on evidence that as little as 10 min of high intensity exercise can improve metabolic health and aerobic capacity [23]. While this is true, recent evidence suggests that low-volume HIIT does not impact body composition measures of total body fat mass, body fat percentage or lean body mass [42]. Thus, a greater volume was probably required to compare body compositional changes between the three different groups.

## 5. Conclusions

In summary, the present study showed that 5:2 IF with or without low volume SIT was effective in reducing body weight and fat mass in individuals with overweight and obesity. However, the combination of both did not further enhance such reductions when compared to IF alone. The benefits on lipid levels and markers of glucose regulation were minimal and likely driven by fasting and/or weight loss. Sprint interval training alone also provided limited effects; however, a clear benefit was an improvement in cardiorespiratory fitness which indicates the workload stress of the training was sufficient to induce such adaptations. Therefore, combining both IF with exercise training is required to obtain benefits in both body composition and cardiometabolic relevant measures. Further research is required to determine the optimal balance between sufficient caloric reduction and/or exercise volume/intensity to induce superior benefits when both are combined, but also ensuring that each strategy is practical to follow and/or time-efficient to support compliance long-term.

## Figures and Tables

**Figure 1 ijerph-19-07969-f001:**
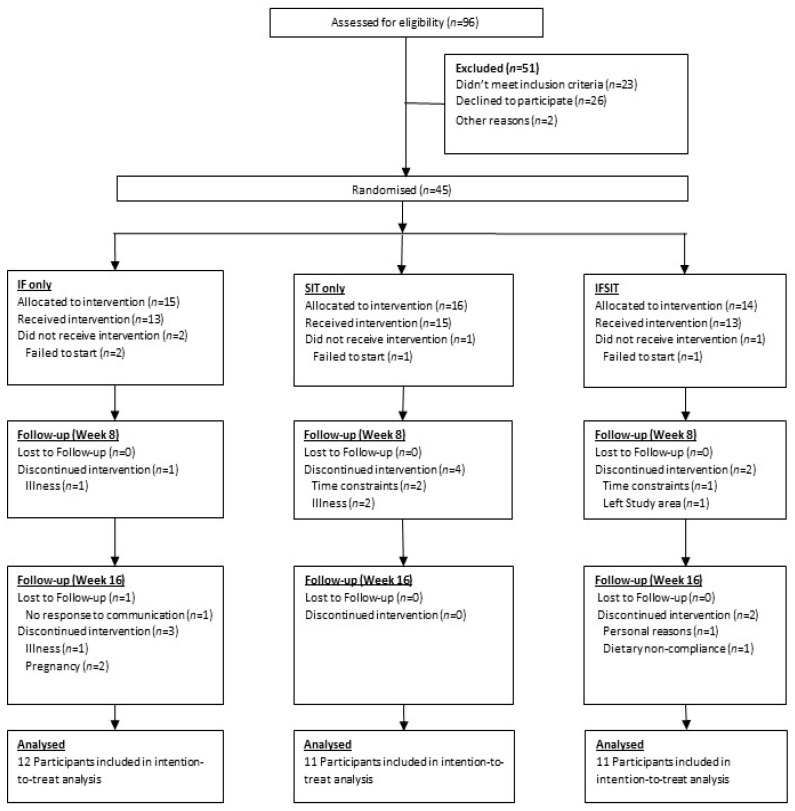
CONSORT diagram summarising the recruitment process, dropouts and randomisation.

**Figure 2 ijerph-19-07969-f002:**
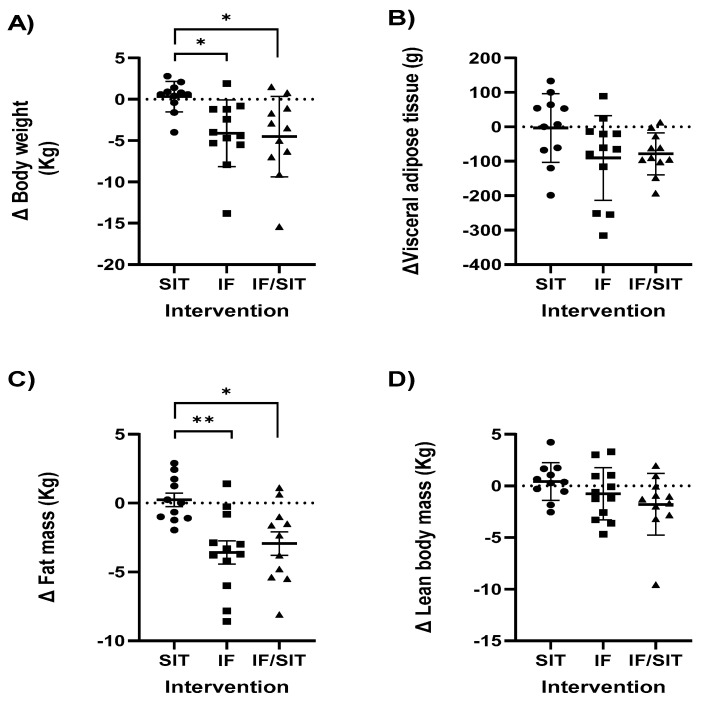
Individual and group (mean ± SD) changes from baseline to post intervention in (**A**) body weight; (**B**) visceral adipose tissue; (**C**) fat mass and (**D**) lean body mass for each diet and/or exercise intervention. * and ** denotes a significant difference between groups (*p* < 0.05 and *p* < 0.01, respectively). Abbreviations: SIT: sprint interval training, IF: intermittent fasting and IF/SIT: intermittent fasting and sprint interval training.

**Figure 3 ijerph-19-07969-f003:**
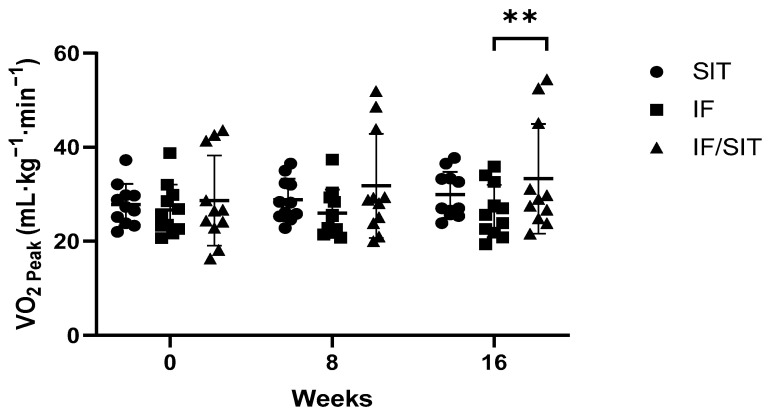
The effect of diet and/or exercise intervention on VO2_peak_. Values are mean ± SD ** denotes a significant difference between the IF and IF/SIT groups (*p* < 0.01). Abbreviations: SIT: sprint interval training, IF: intermittent fasting and IF/SIT: intermitting fasting and sprint interval training.

**Table 1 ijerph-19-07969-t001:** Participant baseline characteristics.

	SIT	IF	IF/SIT
*n*	11	12	11
Age	32 ± 8.3	37 ± 5.9	39 ± 6.8
Sex(M/F)	1/10	2/10	3/8
Height (cm)	168 ± 9.3	164 ± 5.5	167 ± 8.1
Weight (kg)	90 ± 20	82 ± 12	96 ± 8.1
BMI (kg/m^2^)	32 ± 4.4	30 ± 3.9 *	34 ± 2.1 *
Hip Measurement (cm)	114 ± 10.5	112 ± 7.7	118 ± 7.8
Waist Measurement (cm)	101 ± 13	98 ± 13	105 ± 5.3
Visceral Fat Mass (g)	653 ± 256	616 ± 221	681 ± 135
Fat Mass (kg)	34 ± 8.3	31 ± 7.9	36 ± 9.7
Lean Mass (kg)	53 ± 14	47 ± 6.2	56 ± 9.7
Ethnic Origin:			
Aboriginal and Torres Strait Islander	0	0	1
Asian	1	2	0
European	10	10	10
VO_2peak_ (mL·kg^−1^·min^−1^)	27.8 ± 4.4	26.4 ± 5.4	28.7 ± 9.6
VO_2peak_ (L/min)	2.61 ± 0.92	2.02 ± 0.54	2.74 ± 0.95

Note: * denotes a significant difference between groups (*p* < 0.05). Abbreviations: SIT: sprint interval training, IF: intermittent fasting and IF/SIT: intermitting fasting and sprint interval training.

**Table 2 ijerph-19-07969-t002:** The effect of diet and/or exercise intervention on waist circumference and waist to hip ratio.

	SIT		IF		IF/SIT	
Week	Week 0	Week 16	Week 0	Week 16	Week 0	Week 16
	*n* = 11	*n* = 9	*n* = 12	*n* = 8	*n* = 11	*n* = 11
Waist circumference (cm)	100 ± 13	99 ± 13	97 ± 13	94 ± 10	105 ± 8	96 ± 12
Waist to Hip ratio	0.88 ± 0.07	0.87 ± 0.08	0.87 ± 0.08	0.86 ± 0.07	0.89 ± 0.07	0.84 ± 0.08

Values are mean ± SD. Abbreviations: SIT: sprint interval training, IF: intermittent fasting and IF/SIT: intermitting fasting and sprint interval training.

**Table 3 ijerph-19-07969-t003:** The effect of diet and/or exercise intervention on peripheral and central blood pressure and pulse wave velocity.

	SIT		IF		IF/SIT	
Week	Week 0	Week 16	Week 0	Week 16	Week 0	Week 16
	*n* = 11	*n* = 9	*n* = 12	*n* = 8	*n* = 11	*n* = 11
PBP Systolic (mmHg)	130 ± 13	127 ± 13	122 ± 9	123 ± 8	130 ± 12	134 ± 12
PBP Diastolic (mmHg)	80 ± 8	77 ± 10	76 ± 7	77 ± 7	77 ± 7	80 ± 6
CBP Systolic (mmHg)	115 ± 11	116 ± 15	110 ± 7	111 ± 7	116 ± 11	120 ± 11
CBP Diastolic (mmHg)	81 ± 8	78 ± 11	77 ± 7	77 ± 7	78 ± 7	81 ± 6
Pulse wave velocity (m/s)	6.1 ± 1.0	6.3 ± 1.4	5.8 ± 0.9	5.7 ± 0.8	6.6 ± 0.7	6.6 ± 0.9

Values are mean ± SD. Abbreviations: PBP: peripheral blood pressure, CBP: central blood pressure, SIT: sprint interval training, IF: intermittent fasting and IF/SIT: intermitting fasting and sprint interval training.

**Table 4 ijerph-19-07969-t004:** The effect of diet and/or exercise intervention on cardiometabolic biomarkers.

	SIT		IF		IF/SIT	
Week	Week 0	Week 16	Week 0	Week 16	Week 0	Week 16
	*n* = 11	*n* = 9	*n* = 12	*n* = 8	*n* = 11	*n* = 11
Fasting Glucose (mmol/L)	4.9 ± 0.6	4.8 ± 0.7	5.0 ± 0.6	5.0 ± 0.6	4.9 ± 0.4	4.8 ± 0.3
HbA1c (mmol/mol)	32 ± 3.4	32 ± 4.1	33 ± 3.5	33 ± 4.2	31 ± 3.3	32 ± 2.7
Glucose Tolerance (AUC)	777 ± 194	793 ± 158	840 ± 223	769 ± 192	817 ± 169	714 ± 133
HOMA-IR	1.6 ± 0.8	1.8 ± 1.0	1.5 ± 0.5	1.4 ± 0.5	1.7 ± 0.7	1.5 ± 0.6
Total Cholesterol (mmol/L)	4.5 ± 1	4.4 ± 0.9	4.7 ± 1.7	4.6 ± 1.9	4.7 ± 1.0	4.4 ± 1.0
Triglycerides (mmol/L)	1.0 ± 0.5	1.1 ± 0.6	1.2 ± 0.6	1.1 ± 0.6	1.0 ± 0.5	0.9 ± 0.3
LDL Cholesterol (mmol/L)	2.5 ± 0.7	2.4 ± 0.7	2.8 ± 1.0	2.7 ± 1.3	2.9 ± 0.9	2.8 ± 0.9
HDL Cholesterol (mmol/L)	1.4 ± 0.3	1.5 ± 0.4	1.7 ± 0.5	1.7 ± 0.4	1.3 ± 0.3	1.3 ± 0.3

Values are mean ± SD. Abbreviations: AUC: Area under the curve, LDL: low-density lipoprotein, HDL: High-density lipoprotein, SIT: sprint interval training, IF: intermittent fasting and IF/SIT: intermitting fasting and sprint interval training. Please note for glucose tolerance (AUC), *n* = 10 (week 0) and *n* = 9 (week 16).

**Table 5 ijerph-19-07969-t005:** Participants’ self-reported total calorie and macronutrient intakes during the diet and/or exercise intervention.

	SIT			IF			IF/SIT		
Week	0	8	16	0	8	16	0	8	16
	*n* = 7	*n* = 7	*n* = 7	*n* = 6	*n* = 6	*n* = 6	*n* = 7	*n* = 7	*n* = 7
CHO (g·kg·bw per day)	2.14 ± 0.7	1.83 ± 0.8	1.76 ± 0.8	2.20 ± 0.9	2.27 ± 0.4	1.63 ± 1.0	2.22 ± 1.2	1.79 ± 0.60	1.79 ± 0.41
Fat (g·kg·bw per day)	0.63 ± 0.2	0.73 ± 0.4	0.61 ± 0.2	0.73 ± 0.3	0.73 ± 0.2	0.68 ± 0.2	0.88 ± 0.6	0.68 ± 0.28	0.61 ± 0.19
Protein (g·kg·bw per day)	0.83 ± 0.2	0.81 ± 0.3	0.97 ± 0.3	0.81 ± 0.4	1.86 ± 0.7	0.97 ± 0.5	0.89 ± 0.7	0.79 ± 0.59	0.80 ± 0.64
Avg. DCI (kcal)	1710 ± 305	1688 ± 652	1862 ± 902	1594 ± 322	1574 ± 479	1450 ± 267	2181 ± 1057	1698 ± 400	1687 ± 479
DCI (% of predicted RDI)	91 ± 22	83 ± 49	83 ± 10	88 ± 19	86 ± 23	81 ± 10	104 ± 61	83 ± 19	69 ± 36
Avg. Fasting CI (kcal)	n/a	n/a	n/a	463 ± 103	541 ± 84	495 ± 91	543 ± 180	604 ± 171	521 ± 174
CIF (% of predicted RDI)	n/a	n/a	n/a	25 ± 3.1	29.7 ± 2.8	28 ± 4.5	24.7 ± 6.6	29.5 ± 8.2	24.8 ± 7.1

Values are mean ± SD. Abbreviations: CHO: carbohydrate, DCI: daily calorie intake, CI: calorie intake, CIF: caloric intake on fasting days, SIT: sprint interval training, IF: intermittent fasting and IF/SIT: intermitting fasting and sprint interval training Please note that predicted RDI was based on the Mifflin-St. Jeor equation with a low active ‘activity factor’ applied. The calculated DCI values in Table 5 are independent of fasting days.

**Table 6 ijerph-19-07969-t006:** Participants’ self-reported total physical activity and exercise intensity levels during the diet and/or exercise intervention.

	SIT			IF			IF/SIT		
Week	0	8	16	0	8	16	0	8	16
	*n* = 11	*n* = 9	*n* = 8	*n* = 8	*n* = 8	*n* = 8	*n* = 9	*n* = 9	*n* = 9
Total physical activity (MET-min/week)	1722 ± 1046	2780 ± 2539	2136 ± 1616	4611 ± 5410	2388 ± 2224	1980 ± 1621	3964 ± 4459	3933 ± 2924	4552 ± 5503
Intensity (walking/moderate/ vigorous) %	36/41/23	49/48/3	53/47/0	45/43/12	50/41/9	53/47/0	25/39/36	27/55/18	19/40/41

Values are mean ± SD Abbreviations: MET: metabolic equivalent, SIT: sprint interval training, IF: intermittent fasting and IF/SIT: intermitting fasting and sprint interval training.

## Data Availability

The datasets generated during and/or analysed during the current study are available from the corresponding author on reasonable request.

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
