# Peer review of "Effects of Intermittent Energy Restriction Alone and in Combination with Sprint Interval Training on Body Composition and Cardiometabolic Biomarkers in Individuals with Overweight and Obesity"

_ijerph, 2022, doi:10.3390/ijerph19137969_

Round 1

Reviewer 1 Report

Thankyou for the opportunity to review this paper. The authors should be congratulated on a well written manuscript presenting novel findings on the combined impact of intermittent fasting and high intensity interval training. I have some minor comments for your consideration to strengthen the manuscript: 

Introduction

Good introduction, narrative clear.

Line 63-69: repetition of notwithstanding and I don’t think it makes sense in either context

Method:

1. Were there any inclusion/exclusion criteria around current dieting practices or previous experiences with IF etc? this may have had an impact on findings

2. What were the instructions around IF and the 500/600 cal meal that could be eaten on fasting days? Could this be eaten whenever? On non-fasting days was there an instruction to not consume more than typical? I have seen in other IF studies that there has been a tendency to overindulge after fasting

3. Was the timing of the SIT sessions consistent for participants?

Results

Great to see compliant participants!
